# A Fouling Comparison Study of Algal, Bacterial and Humic Organic Matters in Seawater Desalination Pretreatment Using Ceramic UF Membranes

**DOI:** 10.3390/membranes13020234

**Published:** 2023-02-15

**Authors:** Mohammed Al Namazi, Sheng Li, Noreddine Ghaffour, TorOve Leiknes, Gary Amy

**Affiliations:** 1Water Desalination and Reuse Center (WDRC), Biological and Environmental Science and Engineering Division (BESE), King Abdullah University of Science and Technology (KAUST), Thuwal 23955-6900, Saudi Arabia; 2Desalination Technology Research Institute (DTRI), Saline Water Conversion Corporation (SWCC), Al Jubail 31951, Saudi Arabia

**Keywords:** fouling, ceramic UF membrane, seawater desalination, pretreatment

## Abstract

This study investigates three types of organic matter, namely algal organic matter (AOM), bacterial organic matter (BOM), and humic organic matter (HOM). These organics are different in properties and chemical composition. AOM, BOM and HOM were compared in terms of organic content, fouling behavior, and removal efficiency in ceramic UF filtration. UF experiments were conducted at a constant flux mode using 5 kDa and 50 kDa ceramic membranes. Results showed that 5 kDa membrane removed more transparent exopolymer particles (TEP)/organics than 50 kDa membranes, but less fouling formation for all the three types of organic matters tested. Membranes exhibited the lowest trans-membrane pressure (TMP) during the filtration of HOM, most probably due to the high porosity of the HOM cake layer, contributed by big HOM aggregates under Ca bridging effect. AOM shows the highest MFI-UF (modified fouling index-ultrafiltration) and TMP (transmembrane pressure) values among the three organics and during all filtration cycles for both membranes. The AOM fouling layer is well known for having high fouling potential due to its compressibility and compactness which increase the TMP and eventually the MFI values. AOM and BOM organics exhibited a similar fouling behavior and mechanism. Furthermore, the divalent cations such as calcium showed a significant impact on membrane fouling. That is probably because calcium ions made the membranes and organic matter less negatively charged and easier to deposit on membranes, thus, enhancing the membrane fouling significantly.

## 1. Introduction

A study in the intake bay of Al-Jubail area at the site of the SWRO plant intake location revealed excessive growth of filamentous algae, as revealed by biofouling monitoring coupons [1]. The intake point is at the dead end of the bay, where the water turnover is low, and the water column is somewhat stagnant. Although the algae are disinfected by chlorine dosage, they reach the plant structure and create severe filtration problems [1]. The consequence of this is severe reduction in product water output and frequent membrane cleaning due to fouling. Nevertheless, many studies nowadays have revealed that membrane and permeate flux are affected by fouling in general and bio-fouling in particular. In RO system, a positive relationship has been found between Transparent Exo-polymer Particles (TEP) levels in feed water and biofouling rate on the membrane [2,3,4,5]. Furthermore, autopsy analysis of fouled RO membranes showed clearly the existence of alcian blue substances representing TEP on the membrane surfaces. Likewise, other studies (Villacorte et al. 2010 and Bar-Zeev et al. 2012) outlined the occurrence of biofouling associated with TEP in seawater membrane filtration [6,7]. Moreover, the fouling mechanism is strongly linked to TEP source. TEP particles can be derived either from Algae (AOM) or Bacteria (BOM). In both cases, AOM and BOM can be purified or separated from their source. However, TEP cannot purify or separate from AOM and BOM. Therefore, this point is highly considered when a time comes to fouling study. HOM is totally different in terms of molecular structure than AOM and BOM.

For this reason, HOM has been added to study the differences between them within the following hypothesis. Many studies have been carried out to explain the role of TEP in membrane biofouling.

Once the role of TEP in membrane fouling has been asserted, many studies have been carried out to remove TEP from water treatment systems. About two decades ago, UF was discovered to be one of the promising technologies in water industry. Ten years later, UF has proved by itself a wide spectrum of contaminants removal such as viruses, Giardia, and bacteria [8]. Earlier studies showed that particulate TEP (≥0.4 μm) can be easily removed by UF with in-line coagulation pretreatment. Kennedy et al. (2009) concluded that 100% removal of TEP (≥0.4 μm) was achieved after UF [9]. However, this study was only concerned about particulate TEP removal, indicating that colloidal TEP has not been taken into account. Another supportive study by Villacorte et al. (2009) outlined that particulate TEP was readily removed from integrated membrane system (IMS) by typical pretreatments such as low-pressure membranes [10]. They also stated that colloidal TEP, less than 0.1 μm, could not be completely removed from source water by microfiltration (MF) and UF. Furthermore, TEP levels were also reduced by only 30% after a combination of coagulation, sand filtration, and MF [11]. In summary, most previous studies of particulate TEP removal were clearly successful whereas removal of colloidal TEP from various areas of integrated membrane systems remains a challenge. For this reason, it is very likely that these TEP precursors (≤0.4 μ) can reach RO membranes and foul them. These days, ceramic membranes have been gained a high reputation among water production society due to their mechanical, chemical, and thermal stability [12]. These inorganic membranes have also long working life, high flux, and fouling potential compared to organic membranes such as polymeric membranes [13]. Seawater temperature in the Arabian Gulf ranges from 16 °C in winter and 33 °C in summer [14]. Hence, ceramic membranes make it possible because of their resistance to high temperature values and eventually, UF ceramic membranes represent a successful combination for organic fouling pretreatment.

Although they are quite similar in organic fracture, AOM and BOM could behave differently in terms of characterization and fouling mechanism compared to HOM. HOM might exhibit a different fouling behavior and mechanism compared to AOM and BOM due to its small molecular weight cut off or (MWCO) and its ability to interact with divalent cations such as calcium. Smaller LMWCO UF ceramic membranes remove more efficiently TEP/organics than larger ones, will have lower flux but more stable than membranes with larger pore sizes. However, it was not clear yet what size of ceramic membrane is more efficient on removing different organics. Moreover, the fouling mechanism of AOM, BOM, and HOM on ceramic UF membranes is not well understood, and the potential impact of divalent cations on the fouling formation of different organic matter is also not yet systematically investigated.

Therefore, this study was conducted to illustrate the potential effectiveness of ceramic UF membranes on removing different organics in seawater, and its possible influencing factor on fouling formation.

## 2. Materials and Methods

### 2.1. AOM, BOM, and HOM Extraction

To control algal growth, water samples (100 mL) were collected from various cultures every three days. In addition to TOC, DOC, pH, chlorophyll–a, and total cells were monitored using flow cytometry. The AOM extracted at the stationary-decline phase (10–18 days) using the protocol previously reported in literature [15]. The algal cultivation was conducted in an environmental chamber. This chamber is fully equipped with incubation as well as aeration system to maintain the best environmental conditions for biological culturing. The method developed by Myklestad (1995) was applied to cultivate and extract AOM in the lab using a new strain (*Cheatocers affines*, CA) imported from Culture Collection of Algae and Protozoa (CCAP) company, Oban, Scotland [16]. The strain was preserved in the environmental chamber at a constant growth temperature (20 °C) and a light intensity of approximately (50 μmol/m^2^·s) To mimic real day and night intervals, an artificial light control set for 12 h-on/12 h-off was used. The culturing protocol was started with a 2 mL inoculum of new strain in a 50 mL sterile culture tube enriched with marine nutrients based on Guillard F/2 medium, and then incubated in the environmental chamber. The Guillard’s Marine Water Enrichment Solution, from Sigma Aldrich, is enriched in major nutrients required for diatoms cultivations as described by Guillard [17]. A week later, another inoculation in a 1000 flask was carried out by spiking 5–7 mL of 50 mL culture tube into 200 mL of autoclaved and enriched raw seawater (RSW). The 200 mL was then transferred into 10 L autoclaved glass bottles containing 5 L of culture medium after a week of incubation. Finally, the 5 L solution was kept in the environmental chamber at the set conditions mentioned above for 20 days. Extraction of BOM was conducted after 10 days of culturing and extracted according to the protocol reported by Li et al. [15]. A commercial seawater humic substance (SWAN RIVER) was used as HOM model. The HOM is a popular model compounds used in membrane fouling mechanism study, which is isolated from the Suwannee river, and its characteristics has been previously reported [18].

### 2.2. Filtration Experiment Protocol

A bench scale dead-end UF ceramic membrane setup used in this study. The membrane holder is provided by TAMI Company and has a diameter of 45 mm (Section 2.3). The UF ceramic membranes were also supplied by the same company and they have the same diameter for membrane specifications. The operational conditions set for experiments were 18 filtration cycles for 30 min each with synthetic seawater (Section 2.4) and 1-min backwashing using Milli-Q water (Section 2.5). The setup is mainly consisting of 2 gear pumps, for feed and backwash respectively. The pumps connected to feed water tank (2 L) and backwash tank (2 L) containing MQ water. A digital balance from Mettler Toledo Company was used to measure the flux. The setup is automated using a data logger that directly connected to LabVIEW computer software. TMP as a function of flow rate versus time was monitored every 30 s through the computer program. The flux was constant at (241 L/m^2^/h) with constant flow rate at 7 mL/m. At the beginning of filtration run, the initial permeability was tested using MQ water to ensure the standard permeability by membrane manufacture. Furthermore, due to the SEM and TEP visualization destructive analysis, each filtration cycle has used new ceramic membrane.

After experiments, the corresponding water samples (feed, permeate) and used membranes were analyzed using the analysis techniques described in Section 2.6.

### 2.3. Ceramic UF Membranes

The ceramic UF membrane utilized in this study was obtained from TAMI company. The characteristics of the ceramic membranes are shown in Table 1, and clean water permeances of 50 kDa and 5 kDa membranes are 92.6 and 76.9 L/(m^2^·h·bar), respectively.

### 2.4. Feed Water

The feed water was prepared using synthetic seawater quality as presented in (Table 2). The feed water solution was diluted with AOM, BOM, and HOM with total DOC 0.7 mg/L.

### 2.5. Backwash Water

The previous studies revealed Milli-Q water was strongly able to remove NOM fouling from UF membranes [19,20,21,22,23]. While the previous studies were mostly carried out on polymeric membranes and different NOM characteristics, this study used MQ water for determine the role of divalent cations such as calcium on fouling mechanism on ceramic membrane as MQ water is free from any cations.

### 2.6. Membrane and Water Sample Analysis

#### 2.6.1. Scanning Electron Microscope (SEM)

To help studying the fouling mechanism on ceramic membranes, SEM equipped with cryo-stage and cryo preparation chamber has been used for membrane morphology. For cross sectional image, Focused Ion Beam (FIB) was used in combination with SEM. Energy Dispersive X-ray or EDAX was used for elemental analysis. This analysis was carried out at KAUST Core Lab.

#### 2.6.2. AOM/BOM/HOM/TEP Visualization

All the three organic samples have been visualized using alcian blue staining. After each filtration cycle, membrane samples were dried in room temperature and a couple of alcian blue drops were added on the membrane surface and, 10 min later, these samples were visualized using a microscope [15].

#### 2.6.3. AOM and BOM Cultivation

In this experiment, the most dominant algae species present in the Gulf seawater such as Chaetoceros affins (CA) was acquired from the Culture Collection of Algae and Protozoa (CCAP; Oban, Scotland). CA represents the main producer of micro algae biopolymers [15]. All these species belong to the diatom group that represents one of the most common types of phytoplankton dominated in seawater. The cultivation was taken place at the WDRC lab. The cultivation period for AOM extraction to be ready was 20 days. Dominant marine bacteria in the Red Sea water, mainly Pseudidiomarina, atlantica (*P. atlantica*) was cultured at WDRC lab to extract BOM [4,15,24].

#### 2.6.4. TEP Analysis

TEP concentrations in AOM and BOM were measured using a method developed by Li (2015) [15]. Particulate TEP (P-TEP) having higher sizes than 0.4 μm was measured by 0.4 μm pore size polycarbonate membrane (PC) filter (Whatman Nuclepore) whereas colloidal TEP (C-TEP) was measured using 0.1 μm PC.

#### 2.6.5. MFI-UF

This tool is basically developed to measure the membrane fouling potential of the feed water of membrane filtration system. MFI developed by Schippers and Verdouw (1980) [25], then improved by Boerlage et al. (2004) and Salinas Rodriguez et al., (2012) as MFI-UF inspector to quantify membrane fouling [26,27].
MFI = (ηI)/(2ΔP A^2^)
where ΔP is the reference pressure value of 2 bar, η is the reference viscosity at 20 °C and A is the reference filtration area of 13.8 × 10^−4^ m^2^.

MFI-UF is a further developed method, using ultrafiltration membranes at constant flux. ‘I’ is the slope of the linear region in a plot of ΔP versus time and is a characteristic of the cake/gel filtration mechanism.

#### 2.6.6. Particle Size Distribution and Calcium Binding Experiment

The Malvern Nano-sizer equipment has been used for measuring the particle size and the ability of these three organics for binding the calcium molecules. All feed samples; AOM, BOM, and HOM were tested before and after the addition of calcium in order to examine the ability of binding. All samples were tested in duplicate.

#### 2.6.7. LC-OCD Characterization

To characterize the NOM, LC-OCD was developed to identify quantitative information and qualitative results regarding organic compounds in natural water. LC-OCD model 8 system has UV detector (UVD), online organic carbon detector (OCD), and organic nitrogen detector (OND). In this study, the characteristics of different constituents of NOM typically identified are; biopolymer, humics, building blocks, LMW acids, and LMW neutrals. Seawater samples were collected in 20 mL glass vials and filtered with a Whatman filter (pore size = 0.45 μm). Then, compounds were separated using two-column size (250 nm × 20 mm, Toyo pearl TSK HW-50S).

## 3. Results and Discussion

### 3.1. Characteristics of Organics

LC-OCD results (Figure 1) showed the AOM growth during cultivation days. The results show the chromatograms of AOM culture medium. The biopolymer growth started from the exponential phase (day 2 to day 8) to stationary-death phase (day 10–day 18). The biopolymer peaks are clearly detected between 26–38 min retention times. This retention time is within the range described by Huber et al., (2011) [15,18]. The highest biopolymer peak was observed on day 14 when the algal cells counts reached the maximum. Another major peak of the chromatogram can be observed between 45–53 min and represents the low molecular weight (LMW) acids. This LMW acid peak could be attributed to the F/2 culture medium where the EDTA agent is one of its components [2]. Small peaks for building blocks and LMW neutrals were also detected as shown in (Figure 1). The chromatograms of culture medium correlated positively with AOM culture. They show one major peak for LMW acids which appeared in the same retention time range of LMW acids for AOM. Apart from this, no peaks were observed for culture medium during all AOM growth phases. Furthermore, minor peaks of biopolymer, building blocks, humics, and LMW acids and neutrals appeared in the RSW solution. However, the amount of these fractures is very low when they compared to the final AOM concentration (Figure 1). Thus, the RSW fractions have little impact or effect on the final AOM composition. From the perspective of the LC-OCD results, AOM mainly consists of a biopolymer fraction (i.e., polysaccharides and proteins) and some minor concentrations of building blocks and LMW neutrals. A similar biopolymer increase during marine bacteria growth was observed in a previous study as well [4,15].

### 3.2. Filtration Performance of Ceramic UF Membranes

Flux profile presented in (Figure 2) reveals the fouling scenario during UF ceramic filtration experiment as a function of TMP versus time for all organic matters. During the initial period of UF filtration using 50 kDa membranes (A), the TMP increased smoothly and gradually, and then stabled after 5–10 min of filtration cycle. Generally, 50 kDa membranes was more exposed to pore blockage followed by cake layer fouling compared to 5 kDa membranes. This is attributed to the large pore sizes of the 50 kDa membrane that allow the small particles of the three organics particularly HOM to penetrate the pores. In addition, it is probably due to the deformability TEP derived from AOM and BOM particles in the feed water, both leading to pore blockage fouling mechanism. The highest TMP was observed at 5 kDa filtration operation (4.5 bars) whereas the 50 kDa UF ceramic membranes had the highest TMP at 3.5 bars, due to the lower MWCO of the 5 kDa membranes and their corresponding higher membrane resistance. AOM and BOM showed thinner cake layer during all filtration cycles using 5 kDa membranes. As mentioned earlier, TEP particles derived from AOM and BOM have evolved this process by making the cake layer more compact and compressible. Furthermore, some previous studies support this fact which in turns enhance the fouling resistance and reduce the porosity of the cake layer [5,7,28]. On the other hand, HOM cake layer was thicker than those for AOM and BOM. However, this cake layer was found to be more porous and less compact, and this was more likely to occur when HOM particles bound with calcium offering big aggregates and leading to high porosity of the HOM cake layer on the membrane surface, and thus less impact on TMP and MFI values as shown in (Figure 2 and Figure 3).

As shown in (Figure 3), the fouling potential was higher with the 50 kDa membrane compared to the 5 kDa one. The AOM represented the highest MFI-UF value (3200 s/L^2^) whereas HOM gave the lowest value (2000 s/L^2^). This observation suggests that the pore blockage mechanism is more dominant for the 50 kDa UF membranes and cake layer of the 5 kDa membrane is most probably dominating the fouling mechanism. In fact, these results are consistent with other studies (Li et al., 2011) [29]. The possibility of the three organic particles to be trapped into the larger membrane pores was higher during UF filtration using 50 kDa membranes, which explains the higher values of MFI-UF. AOM shows the highest MFI-UF values among the three organics and during all filtration cycles for both membranes. The AOM fouling layer is well known for having high fouling potential due to its compressibility and compactness, which increase the TMP and eventually the MFI-UF values [28].

As shown in Figure 4, AOM and BOM showed the highest TEP concentrations in ceramic membrane filtration experiments for both 5 and 50 kDa membranes, which are in agreement with literature as AOM and BOM contain more polysaccharides and eventually more TEP. However, TEP on AOM was higher than BOM because of higher biopolymer fraction in AOM compared to BOM. Regarding TEP removal, this study clearly revealed that UF 5 kDa membranes can remove more TEP/organics (66–80%) compared to the 50 kDa membrane (53–57%) due to their lower MWCO (Figure 4). The results from the ceramic UF membranes were better than those reported before with existing pretreatment DMF and previous reported polymeric UF, which were about 10% and 40%, respectively [3,5].

The particle size distribution results presented in (Figure 5) showed that the smallest particle sizes were HOM, then BOM, and AOM, respectively. However, HOM shows the smallest size before addition of calcium and the largest size after binding with calcium molecules. The HOM particles were in the range of 8–13 nm, suggesting that they can penetrate easily the 50 kDa membrane pores. The study also shows pore blockage with AOM and BOM samples that are larger in terms of particles sizes than the 50 kDa membrane pores. The reason behind this is most likely that the AOM and BOM contain mainly deformable TEP particles that can penetrate the pores structures [30]. In addition, the particle size distributions presented in (Figure 5) show only the average values (mean size), therefore the smaller particles that not in the range presented with particle size distribution from AOM and BOM could pass the 50 kDa UF membranes and cause pore blocking. For 5 kDa membrane, most of organic particles have been rejected probably because of the lower MWCO or this membrane. For 50 kDa membranes, AOM and BOM showed the highest TEP concentrations, which are in agreement with literature as AOM and BOM contain more polysaccharides and eventually more TEP. However, TEP on AOM was higher than BOM because of higher biopolymer fraction in AOM compared to BOM. Regarding TEP removal, the study clearly revealed that UF 5 kDa membranes can remove more TEP/organics compared to the 50 kDa membrane due to their lower MWCO.

The AOM fouling layer came at last with very high organic content/TEP and less calcium concentration. It is well known that CA species can produce more organic matter rather than BOM and HOM and this is most probably the reason behind this phenomenon. Further, the AOM fouling layer was gel-like layer and sticky due to its content of polysaccharides. All these fouling layers of AOM, BOM, and HOM were porous. However, the HOM cake layer was found to be the highest porous layer compared to AOM and BOM layers because of the role of calcium binding that enhances the size of HOM particles, allowing them to create larger particles, porous and less compact cake layer as clearly indicated in TMP and MFI-UF results.

### 3.3. Impact of Calcium on fouling 

Comparative results of calcium concentration deposited on the surface of UF ceramic membranes shown in EDAX elemental analysis presented in (Figure 6). By comparing the two membranes used, the calcium concentration was found to be higher on 50 kDa than 5 kDa membrane surface. Furthermore, the fouling layer of HOM on 50 kDa membrane was found to be the highest compared to AOM and BOM, resulting in a high binding between humic substances and calcium molecules. These findings confirm that 5 kDa membranes were able to remove more organics than 50 kDa membrane. Thus, the concentrations of the three organics particularly humics were observed low on 5 kDa membrane surface, resulting in decreased adsorption between HOM and calcium molecules. However, the adsorption of HOM and the calcium molecules for the 5 kDa membrane was also higher than AOM and BOM due to high calcium concentration deposited on the membrane surface (Figure 6).

## 4. Conclusions

The following conclusions can be drawn from this study.

AOM gave the highest MFI-UF and TMP values among the three organics and during all filtration cycles for both membranes. The AOM fouling layer is well known for having high fouling potential due to its compressibility and compactness which increase the TMP and eventually the MFI-UF values. AOM and BOM organics presented a similar fouling behavior and mechanism. However, AOM was significantly higher compared to BOM in terms of TEP concentrations and gel-like formation. This is probably attributed to the high polysaccharide concentration in AOM.UF 5 kDa membranes can remove more TEP/organics compared to the 50 kDa membranes due to their lower MWCO. A cake layer fouling while the 50 kDa membrane showed a blockage fouling mechanism followed by a cake layer formation. For 5 kDa membranes, AOM and BOM showed thinner cake layer during all filtration cycles as TEP particles derived from AOM and BOM have evolved this process by making the cake layer more compact and compressible which in turns enhance the fouling resistance and reduce the porosity of the cake layer. HOM cake layer was thicker than those for AOM and BOM. This cake layer was found to be more porous and less compact, and this occurred more likely when HOM particles bind with Ca molecules offering big aggregates and leading to high porosity of the HOM cake layer on the membrane surface.The divalent cations such as calcium revealed a strong influence on membrane fouling. In this experiment, the HOM particles were most likely influenced by this phenomenon which bridge/adsorb more organic molecules when interacting with calcium ions making the membrane less negatively charged and enhancing the membrane fouling. However, this fouling was less severe compared to AOM and BOM fouling.

## Figures and Tables

**Figure 1 membranes-13-00234-f001:**
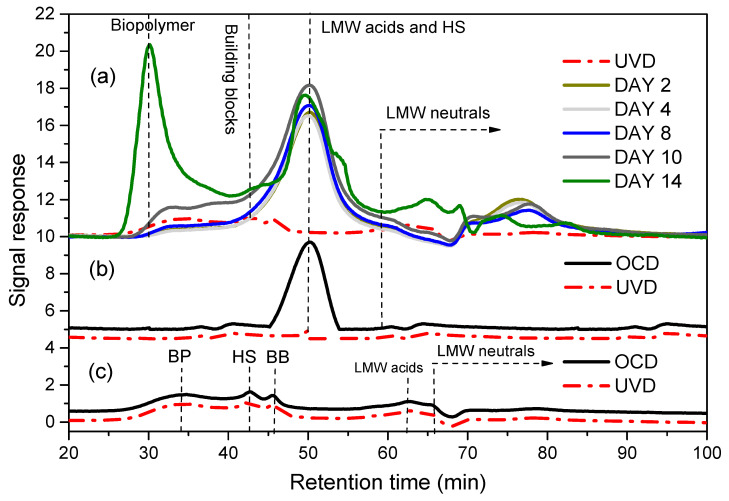
LC-OCD chromatograms for AOM during cultivation (**a**), algae cultivation medium (**b**), and background seawater (**c**).

**Figure 2 membranes-13-00234-f002:**
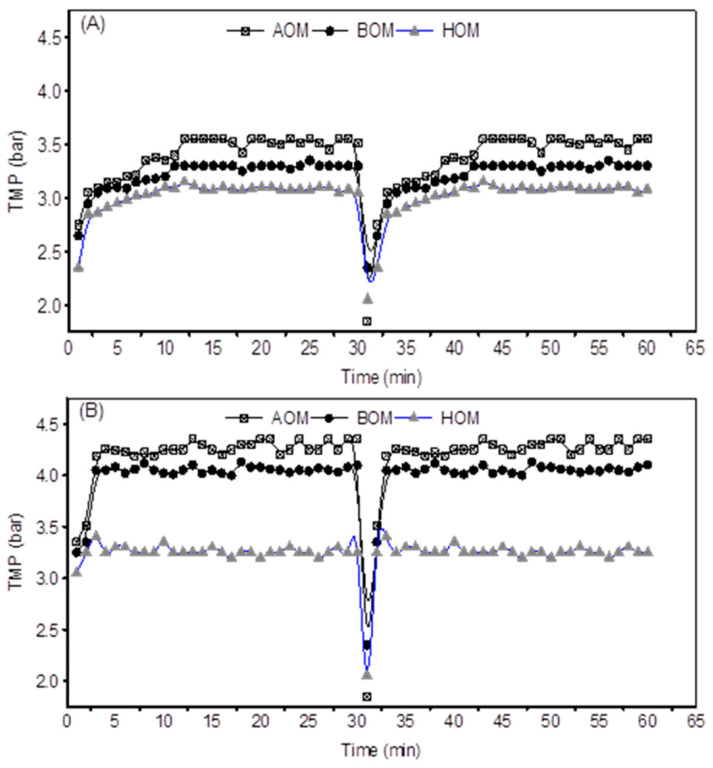
TMP comparison for 50 kDa (**A**) and 5 kDa (**B**). Each filtration experiment was duplicated, and the variation of two experimental data sets are within 10%.

**Figure 3 membranes-13-00234-f003:**
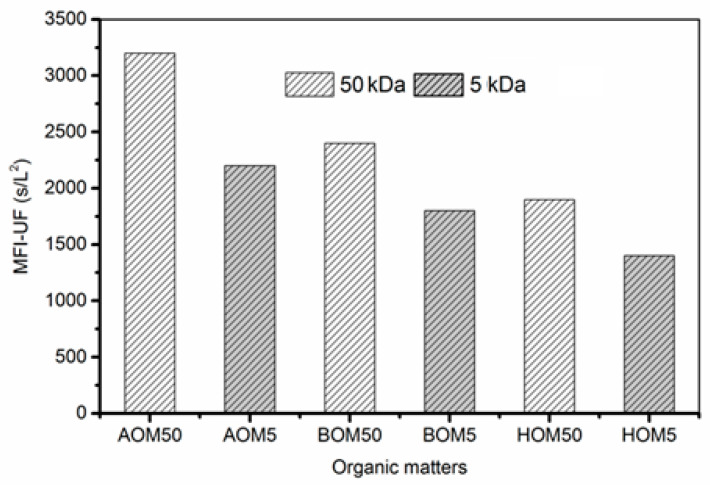
MFI-UF for 50 kDa and 5 kDa ceramic membranes. The MFI-UF were calculated twice for each experiment, and the deviation between two times calculations were within 5%.

**Figure 4 membranes-13-00234-f004:**
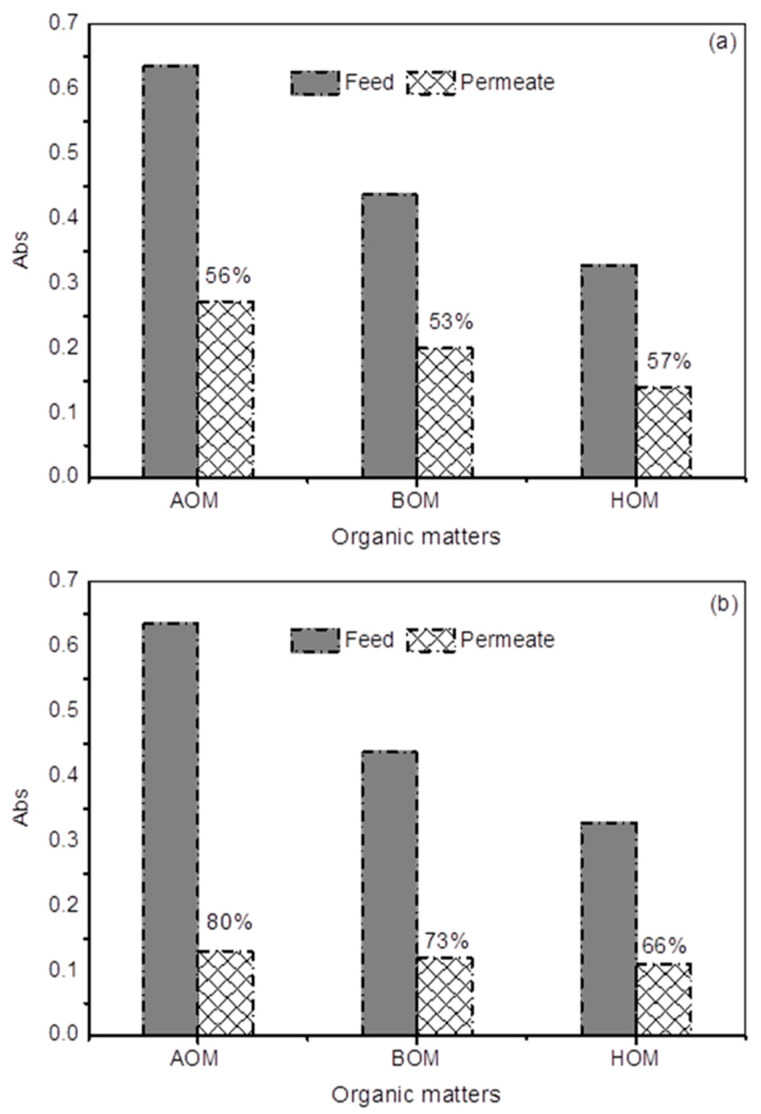
TEP removal as a function of absorbance of (**a**) 50 kDa and (**b**) 5 kDa for all organic matters.

**Figure 5 membranes-13-00234-f005:**
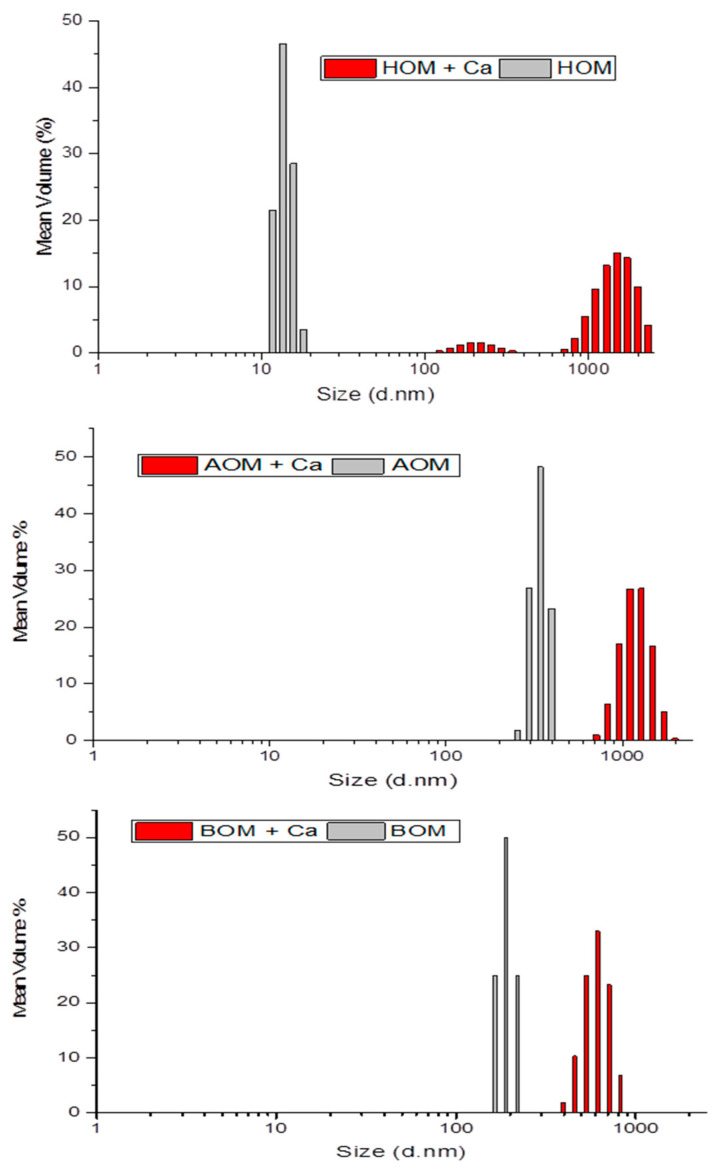
Size distributions of feed water before and after addition of Ca for all types of organic matters. HOM shows the smallest size before addition of calcium and the largest size after binding with calcium molecules.

**Figure 6 membranes-13-00234-f006:**
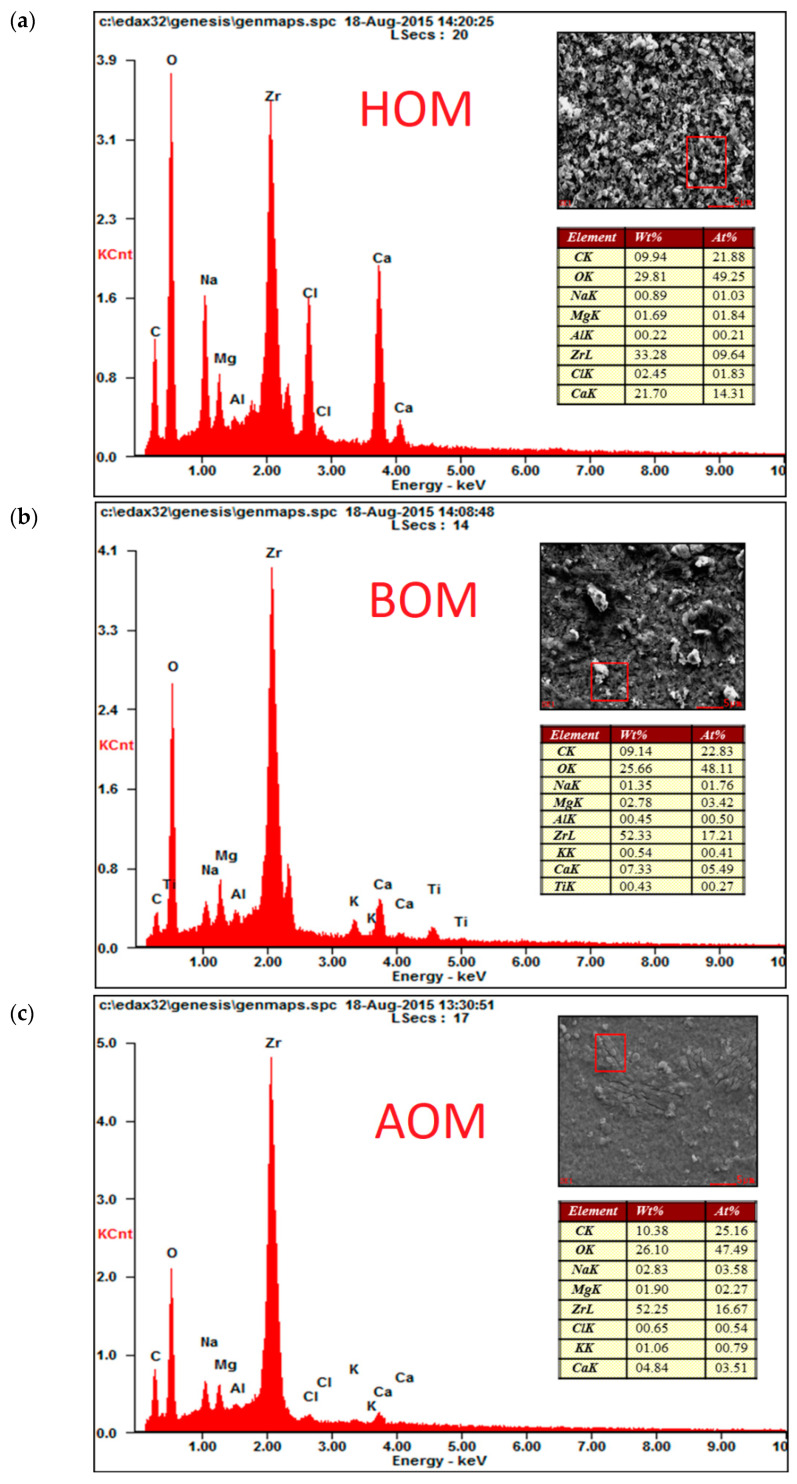
EDAX elemental analysis for (**a**) HOM, (**b**) AOM, and (**c**) BOM in 5 kDa UF membrane filtration.

**Table 1 membranes-13-00234-t001:** Water quality of synthetic seawater used in this study.

Manufacturer	TAMI
Pore size, or MWCO	50 kDa, 5 kDa
Materials	Support layer: TiO_2_ Active layer ZrO_2_ + TiO_2_
Surface Area (cm^2^)	17.4

**Table 2 membranes-13-00234-t002:** Water quality of synthetic seawater used in this study.

Parameters	Concentration mg/L
Chloride (Cl)	19,290 mg/L
Sodium	10,780 mg/L
Boron	5.6 mg/L
Sulfate	2660 mg/L
Potassium	420 mg/L
Calcium	400 mg/L
Magnesium (Mg) DOC mg/L	1320 mg/L 0.7

## Data Availability

Data is unavailable due to company restriction.

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
