# Peer review of "A Fouling Comparison Study of Algal, Bacterial and Humic Organic Matters in Seawater Desalination Pretreatment Using Ceramic UF Membranes"

_membranes, 2023, doi:10.3390/membranes13020234_

Round 1
Reviewer 1 Report
This manuscript entitled “A Fouling comparison study of algal, bacterial, and humic organic matters in seawater desalination pretreatment using UF membranes”, by Mohammed Al Namazi et al., presents the fouling behavior of three types of organic matters in ultrafiltration process using 5 kDa and 50 kDa ceramic UF membranes. It’s an interesting topic. However, the correction and improvement are needed.
Q1 The author presents several meaningful questions in the introduction (Line 77~81). However, some of their answers lack necessary experimental data, such as “ What kind of fouling might have, irreversible or reversible and why? How efficiency of MQ backwashing eliminating these foulants and restore the flux?”
Q2 The relevant information of two UF ceramic membranes should be characterized, such as the water permeance and the clean surface morphology.
Q3 The corresponding calculation for MFI-UF should also be briefly introduced
Q4 How to determine the thickness of foulants cake layer? (Line 220~228, and Line 275~278)
Fig.2: the scales on the axis Y did not correspond to the numbers.
Fig.3: the data lacked error bars
Fig.6: the image was unsharpness.
Typo error:
Line 56 “Girdia” → Giardia
Line 252 “50 KDa” → 50 kDa
Line 303 “UF 5 kD” → UF 5 kDa
Author Response
1. The author presents several meaningful questions in the introduction (Line 77~81). However, some of their answers lack necessary experimental data, such as “ What kind of fouling might have, irreversible or reversible and why? How efficiency of MQ backwashing eliminating these foulants and restore the flux?”
Response: Thanks for the reviewers' comments. The introduction part has been revised in the new manuscript, and the non-discussed research questions has been deleted.
2. The relevant information of two UF ceramic membranes should be characterized, such as the water permeance and the clean surface morphology.
Response: Thanks for the reviewers' comments. The characteristics of used ceramic membranes have been added in the material and method section 2.3 of revised manuscript.
3. The corresponding calculation for MFI-UF should also be briefly introduced
Response: Thanks for the comments. The procedure for MFI-UF calculation is described in detail in cited literatures. To give a brief introduction, the following description has been added to the revised manuscript.
MFI = (ηI) / (2ΔP A2)
where ΔP is the reference pressure value of 2 bar, η is the reference viscosity at 20 °C and A is the reference filtration area of 13.8 10−4 m2.
MFI-UF is a further developed method, using ultrafiltration membranes at constant flux (Boerlage et al., 2004). ‘I’ is the slope of the linear region in a plot of ΔP versus time and is a characteristic of the cake/gel filtration mechanism.
4. How to determine the thickness of foulants cake layer? (Line 220~228, and Line 275~278)
Response: Thanks for the reviewer's comment. The thickness of foulant layer was measured by using the ruler in SEM imagining.
5. Fig.2: the scales on the axis Y did not correspond to the numbers.
Response: Thanks reviewer for the comment. It has been corrected.
6. Fig.3: the data lacked error bars
Response: Thanks reviewer for the comments. The MFI-UF were calculated twice for each experiment, and the deviation between two times calculations were within 5%. This clarification has been added in the revised manuscript.
7. Fig.6: the image was unsharpness.
Response: Thanks reviewer for the comment. Figure 6 was the figure generated by the analysis machine automatically.
8. Line 56 “Girdia” → Giardia
Response: Thanks reviewer for the comment. It has been corrected.
9. Line 252 “50 KDa” → 50 kDa
Response: Thanks reviewer for the comment. It has been corrected.
10. Line 303 “UF 5 kD” → UF 5 kDa
Response: Thanks reviewer for the comment. It has been corrected.
Reviewer 2 Report
This study is conducted to investigate the fouling phenomena for different organic matters including algal organic matter (AOM), bacterial organic matter (BOM), and humic organic matter (HOM). With different properties and chemical compositions, those organic matters are compared regarding organic content, fouling behavior, and removal efficiency in ultrafiltration (UF) filtration process. Two different ceramic UF membranes were utilized, which are with MWCO as 5 kDa and 50 kDa. It was revealed that AOM mainly consists of a biopolymer fraction and other matters. Also, AOM exhibited the highest MFI-UF and TEP values due to high fouling potential compared to other organic matters. Lastly, divalent cations particularly calcium ions promoted membrane fouling by associating with organic matters, and HOM cake layer was thicker due to binding with calcium ions.
The research topic is interesting to understand the phenomena of fouling behaviors of different organic fouling. However, the motivation of research and the purpose of study are not clearly provided, and thus data do not suggest a clear idea of the research. Also, the structure and description should be improved further to meet the academic writing standards. In this sense, this manuscript should be revised significantly to be published in membranes. Major comments can be found below:
Comments:
1. Introduction: The introduction should be clearly restructured including the background, motivation of research, the purpose of study, and clear research direction. Such fundamental components are not clearly stated. Consider general 5-6 paragraphs which consist of more than 6 sentences each. Sentences in line 77-81 are not in an academic style.
2. Fig. 1: LC-OCD for AOM growth is only provided. Provide the reasons why those for BOM and HOM are excluded.
3. Fig. 2: Error bars are not included in the graph. Flux decline is not observed during the filtration period, and it is difficult to claim the occurrence of fouling. Discuss why TMP for HOM is 1 bar lower than AOM and BOM.
4. Fig. 4: The removal rate of ceramic UF for organic matters is not superior. This will lead to the performance decline to the following SWRO process. The authors can state whether this removal rate is appropriate as a pretreatment process for SWRO.
5. Title: It is better to mention “ceramic ultrafiltration membrane” as polymeric membrane is dominant in the current SWRO process.
6. Abstract: The purpose of study is not clearly stated.
7. Material and methods: It is better to integrate sections (e.g., lab-scale experiment, TEP analysis, fouling potential analysis, etc.) Also, AOM, BOM, and HOM details should be clearly provided as a table.
8. Results and discussion: Device the body into several sections with adequate titles.
9. Conclusion: Integrate several paragraphs and present a solid conclusion paragraph.
10. Line 48-52: HOM is utilized due to different characteristics compared to AOM and BOM. It is required to provide more reasonable rationales to use HOM in fouling study (i.e., increase of HOM in seawater, problems in UF pretreatment due to HOM, etc.)
11. Line 73-74: The seawater conditions in the Arabian Gulf are presented without clear references.
12. Line 73-76: Currently, polymeric UF membranes are dominant in SWRO desalination plants in the Arabian Gulf. The advantages of using ceramic membranes should be clearly summarized by comparing those of using polymeric membranes.
13. Line 107-108: Is DOC for AOM, BOM and HOM 0.7 mg/L respectively?
14. UF membranes: More specific information should be provided for ceramic UF membranes in the experiment.
15. Proofread: The manuscript should be proofread by professional editors due to lack of proficiency in academic English. Abbreviations should be defined for the first use of terminations in the manuscript. Superscripts and subscripts should be properly used.
Author Response
1. Introduction: The introduction should be clearly restructured including the background, motivation of research, the purpose of study, and clear research direction. Such fundamental components are not clearly stated. Consider general 5-6 paragraphs which consist of more than 6 sentences each. Sentences in line 77-81 are not in an academic style.
Response: Thanks for the reviewer's comment. The introduction has been restructured accordingly in the revised manuscript.
2. Fig. 1: LC-OCD for AOM growth is only provided. Provide the reasons why those for BOM and HOM are excluded.
Response: Thanks for the reviewer's comment. The BOM growth has been reported in a previous study, and the citation has been added in the revised manuscript. Regarding HOM, it is humic substances isolated from river, and thus there is no growth figure for that.
3. Fig. 2: Error bars are not included in the graph. Flux decline is not observed during the filtration period, and it is difficult to claim the occurrence of fouling. Discuss why TMP for HOM is 1 bar lower than AOM and BOM.
Response: Thanks for the reviewer's comment. The error bar was not included because it was continuous recorded data by PC, and thus difficult to put two duplicated experimental data in one figure, but the variation of duplicate experiments is within 10%. The authors agree with the reviewer that the TMP was stable during the filtration period, and it is difficult to claim the fouling. Description about fouling has been modified accordingly in the revised manuscript. Regarding the reason why HOM exhibit 1 bar lower than others, it has been discussed in section 3.3 of the revised manuscript.
4. Fig. 4: The removal rate of ceramic UF for organic matters is not superior. This will lead to the performance decline to the following SWRO process. The authors can state whether this removal rate is appropriate as a pretreatment process for SWRO.
Response: Thanks for the reviewer's comment. The removal rate of organics in Figure 4 was targeting the TEP removal, which was the particulate and colloidal forms of transparent exopolymer particles. The results from the ceramic UF membranes are comparable and better than those reported before with existing pretreatment DMF and previous reported polymeric UF.
5. Title: It is better to mention “ceramic ultrafiltration membrane” as polymeric membrane is dominant in the current SWRO process.
Response: Thanks for the reviewer's comment. The title has been corrected accordingly.
6. Abstract: The purpose of study is not clearly stated.
Response: Thanks for the reviewer's comment. The purpose of study has been clarified accordingly in the revised manuscript.
7. Material and methods: It is better to integrate sections (e.g., lab-scale experiment, TEP analysis, fouling potential analysis, etc.) Also, AOM, BOM, and HOM details should be clearly provided as a table.
Response: Thanks for the reviewer's comment. The "material and methods" has been restructured accordingly in the revised manuscript.
8. Results and discussion: Device the body into several sections with adequate titles.
Response: Thanks for the reviewer's comment. The "Results and discussion" has been restructured accordingly in the revised manuscript.
9. Conclusion: Integrate several paragraphs and present a solid conclusion paragraph.
Response: Thanks for the reviewer's comment. The "conclusion" has been restructured accordingly in the revised manuscript.
10. Line 48-52: HOM is utilized due to different characteristics compared to AOM and BOM. It is required to provide more reasonable rationales to use HOM in fouling study (i.e., increase of HOM in seawater, problems in UF pretreatment due to HOM, etc.)
Response: Thanks for the reviewer's comment. The HOM is a popular model compounds used in membrane fouling mechanism study, which is isolated from swuanee river. The reason of using HOM has been emphasized in the revised manuscript.
11. Line 73-74: The seawater conditions in the Arabian Gulf are presented without clear references.
Response: Thanks for the reviewer's comment. A clear Arabian Gulf seawater condition reference has been added accordingly in the revised manuscript.
12. Line 73-76: Currently, polymeric UF membranes are dominant in SWRO desalination plants in the Arabian Gulf. The advantages of using ceramic membranes should be clearly summarized by comparing those of using polymeric membranes.
Response: Thanks for the reviewer's comment. The advantages of ceramic UF membranes has been mentioned in the introduction of revised manuscript as below.
"Seawater temperature in the Arabian Gulf ranges from 16 0C in winter and 33 0C in summer. Hence, ceramic membranes make it possible because of their resistance to high temperature values and eventually, UF ceramic membranes represent a successful combination for organic fouling pretreatment"
13. Line 107-108: Is DOC for AOM, BOM and HOM 0.7 mg/L respectively?
Response: Yes, DOC for AOM, BOM and HOM 0.7 mg/L respectively.
14. UF membranes: More specific information should be provided for ceramic UF membranes in the experiment.
Response: Thanks for the reviewer's comment. The information of the used ceramic UF membrane has been listed in the material and methods section.
15. Proofread: The manuscript should be proofread by professional editors due to lack of proficiency in academic English. Abbreviations should be defined for the first use of terminations in the manuscript. Superscripts and subscripts should be properly used.
Response: Thanks for the reviewer's comment. The revised manuscript has been double checked in terms of English.
Round 2
Reviewer 1 Report
I have no more questions.
Reviewer 2 Report
The comments are well addressed. The manuscript is now ready to be published.